# Baseline Normative and Test–Retest Reliability Data for Sideline Concussion Assessment Measures in Youth

**DOI:** 10.3390/diagnostics14151661

**Published:** 2024-08-01

**Authors:** Jennifer V. Wethe, Jamie Bogle, David W. Dodick, Marci D. Howard, Amanda Rach Gould, Richard J. Butterfield, Matthew R. Buras, Jennifer Adler, Alexandra Talaber, David Soma, Amaal J. Starling

**Affiliations:** 1Department of Psychiatry and Psychology, Mayo Clinic School of Medicine and Science, Scottsdale, AZ 85259, USA; 2Department of Otolaryngology-Head and Neck Surgery, Mayo Clinic School of Medicine and Science, Scottsdale, AZ 85259, USA; 3Department of Neurology, Mayo Clinic School of Medicine and Science, Scottsdale, AZ 85259, USA; 4Division of Biostatistics and Clinical Trials, Department of Quantitative Health Sciences, Scottsdale, AZ 85259, USA; 5King–Devick Technologies, Chicago, IL 60522, USA; 6Department of Pediatric and Adolescent Medicine, Department of Orthopedic Surgery, Mayo Clinic School of Medicine and Science, Rochester, MN 55905, USA

**Keywords:** child, adolescent, athletes, brain concussion, reproducibility of results

## Abstract

Tools used for the identification, evaluation, and monitoring of concussion have not been sufficiently studied in youth or real-world settings. Normative and reliability data on sideline concussion assessment measures in the youth athlete population is needed. Pre-season normative data for 515 athletes (93.5% male) aged 5 to 16 on the Standardized Assessment of Concussion (SAC/SAC-Child), modified Balance Errors Scoring System (mBESS), Timed Tandem Gait (TTG), and the King–Devick Test (KDT) are provided. A total of 212 non-injured athletes repeated the measures post-season to assess test–retest reliability. Mean performance on the SAC-C, mBESS, TTG, and KDT tended to improve with age. KDT was the only measure that demonstrated good to excellent stability across age ranges (ICC = 0.758 to 0.941). Concentration was the only SAC/SAC-C subtest to demonstrate moderate test–retest stability (ICC = 0.503 to 0.706). TTG demonstrated moderate to good (ICC = 0.666 to 0.811) reliability. mBESS demonstrated poor to moderate reliability (ICC = −0.309 to 0.651). Commonly used measures of concussion vary regarding test–retest reliability in youth. The data support the use of at least annual sport concussion baseline assessments in the pediatric population to account for the evolution in performance as the child ages. Understanding the variation in the stability and the evolution of baseline performance will enable improved identification of possible injury.

## 1. Introduction

Concussion bears the risk of long-term health consequences, particularly in youth. Most assessment tools used for the screening, evaluation, and monitoring of concussion in older athletes have not been sufficiently studied in youth or in distracting real-world settings that mirror the environment in which the sport is played. Recent research suggests that the rates of concussion in youth sports are higher than previously estimated with an incidence of 5% per season [1]. This highlights the need for improved concussion identification in youth sports and the validation of screening tools in the pediatric population to aid identification.

Test–retest reliability, or stability, indicates the degree to which scores on the battery remain stable over time [2] and is an important factor in determining the validity and usefulness of a concussion-screening tool. Without established reliability, it is difficult to determine whether a decline in test performance is due to low reliability or a result of cognitive impairment due to a concussion. It has been suggested that a reliability of close to 0.90 is necessary for individual decision-making in a baseline assessment model [2,3].

The Sport Concussion Assessment Tool (SCAT) [4,5,6] is a collection of previously published tools for the assessment of cognition, balance, coordination, and signs and symptoms of concussion. It is designed as a rapid assessment administered on the sideline to assist in making return- versus remove-from-play decisions. The SCAT3 [5] assesses cognition via the Standardized Assessment of Concussion (SAC) [7], a brief mental status examination emphasizing orientation, memory, and concentration. Balance and coordination are assessed via the modified Balance Error Scoring System (mBESS) and Timed Tandem Gait (TTG). A detailed concussion symptom rating is included. Most research validating the SCAT has been conducted with high school, college, and professional athletes [5,8,9,10,11,12].

The pediatric version of the Sport Concussion Assessment Tool (Child-SCAT3) [13] was introduced in 2012. In 2017, Nelson and colleagues published preliminary normative data on the Child-SCAT3 components based on 155, 5–13-year-old football and soccer players [14]. Test–retest reliability data were included on a subsample of 57, 9–13-year-old athletes. They found fair stability for the SAC-C total score (*r* = 0.50) and TTG (*r* = 0.46) but notably poor stability for the mBESS-C (*r* = 0.02). Brooks et al. (2017) published normative data on the Child-SCAT3 components using a larger sample of 478, 5–13-year-old athletes, but no test–retest reliability data were provided [15]. Preliminary normative data on Child-SCAT3 components have been published [14,15,16]. Normative studies have found significant age effects with younger athletes displaying weaker performance than older athletes on cognitive and balance measures [14,15,16]. In 2017, the SCAT5/SCAT5-C were introduced [4,17]. The SAC/SAC-C were modified to allow the option of assessing memory via a 10, rather than 5, word list to minimize ceiling effects [4,17]. The Child version now omits the four Orientation questions. Tandem gait is included as part of a Neurological Screen and is simply scored as normal or not. The mBESS is retained for the Balance Examination. A question related to neck pain was added to the Child and Parent Symptom Reports, such that they both contain 21 possible symptoms. The remaining 20 symptoms are the same as the SCAT3-C. Thus, most of the elements of the SCAT3/SCAT3-C were retained in, and are easily translatable to, the SCAT5/SCAT5-C [4,17]. With introduction of the SCAT6/SCAT6-C in 2023, use of a 10-word list for memory assessment became routine [18,19]. Use of TTG and mBESS were retained with optional enhancements to TTG to increase complexity.

Systematic reviews recommend adding a measure of oculomotor function to the SCAT3 as neural pathways are heavily involved in vision and eye movement and oculomotor function appears to be an objective marker of brain injury [20,21]. Thus, we include the King–Devick Test (KDT) (Oak Brook, IL, USA), a timed, rapid number naming test that screens oculomotor function but also incorporates aspects of attention and language. The test can be administered via spiral-bound cards or an electronic tablet application, with tablet administration tending to result in slightly slower test times than card administration [22]. KDT times have been noted to improve with age over the youth through adolescent years [21]. In the meta-analysis, KDT demonstrated excellent test–retest reliability (ICC = 0.92). KDT was also shown to be reliable over a one-year period between tests, modalities (tablet versus cards), and years in 3248 intercollegiate student-athletes, with ICCs ranging from 0.827 to 0.888 [23].

Our objective is to add to the limited pediatric normative data for the SCAT3/SCAT5 components and to provide much needed reliability data on sideline measures for children and adolescents.

## 2. Materials and Methods

In this prospective, multi-year study of concussion in youth sports, 517 American football, ice hockey, and cheer athletes aged 5–16 years completed the KDT, SAC/SAC-C, TTG, and/or mBESS pre-season as part of a larger study conducted in Arizona and Minnesota, USA (Table 1). Athletes were excluded from analysis if they or their parents were unable to speak and understand instructions in English (*n* = 20), reported a concussion within the last month (*n* = 3), failed to report their age (*n* = 2), or were administered the incorrect version of the SAC/SAC-C (*n* = 3), leaving 489 (93.5% male) unique athletes. Twenty-six athletes completed assessments in multiple years, resulting in a final sample of 515 baseline assessments (Figure 1).

A total of 231 athletes repeated the measures post-season. The time between baseline and post-season testing was typically 2.5 to 3 months for American football and cheer and 5 months for ice hockey due to differences in season length. Athletes with a suspected (*n* = 6) or diagnosed (*n* = 21) concussion during the season were excluded from the post-season dataset. One additional athlete was excluded due to having undergone medical sedation the day of the post-season testing. A parent/guardian of each participant provided written informed consent, and all participants provided assent. The study was approved by the Mayo Clinic Institutional Review Board (IRB #14-002811).

### 2.1. Procedure

Participants completed pre-season testing prior to the first game of the season. Post-season testing occurred within 3 weeks of their final game of the season. A multidisciplinary team of medical professionals (e.g., neurologists, neuropsychologists, vestibular audiologist, athletic trainer, medical residents) and research assistants conducted the testing at practice fields/rinks or available facilities (school gyms, conference rooms, hallways), thus distracting environments that mirrored real-world environments of youth sports.

### 2.2. Measures

#### 2.2.1. Standardized Assessment of Concussion (SAC) and Child SAC (SAC-C)

The SAC is a cognitive screening test of the SCAT3 that consists of 4 sections: Orientation (maximum score = 5 for SAC, 4 for SAC-C), Immediate Memory (5-word list practiced 3 times; maximum score = 15), Concentration (maximum score = 5 for SAC, 6 for SAC-C), and Delayed Recall (maximum score = 5). A maximum total score of 30 is generated by adding the 4 sub scores from each section. Memory and Concentration: Digits Backward both provide alternate lists to minimize practice effects. In the current study, the lists utilized for each athlete were chosen via a quasi-random method at both baseline and post-season; thus, athletes were tested on different memory lists at each test session.

#### 2.2.2. Modified Balance Errors Scoring System (mBESS)

During mBESS, subjects balance during three, 20 s stances [double leg (DL), single leg (SL), tandem (TS)] with eyes closed on a firm surface. Errors (e.g., stumbling, lifting hand off of the iliac crest) are recorded with a maximum of 10 errors for each stance. For SCAT3-C, athletes under age 13 years are only scored on DL and TS, not SL; however, athletes in our sample completed all three stances at baseline, regardless of age.

#### 2.2.3. Timed Tandem Gait (TTG)

TTG requires the subject to quickly and accurately walk a 3 m tape, turn 180°, and return to the starting point all while using a tandem (heal-to-toe) gait without shoes. Patients are asked to complete this as quickly and accurately as possible. Four trials are completed with the fastest time serving as the final score.

#### 2.2.4. King–Devick Test (KDT)

The KDT is a test of saccadic eye function in which participants read numbers aloud from left to right across three test cards of increasing difficulty. Time and errors are tracked. KDT time is the cumulative time taken to read all three test cards. At least two test trials are completed pre-season, with the fastest error-free trial serving as the baseline score. The KDT is administered via tablet application on a standard-sized iPad or Android tablet or via 6 × 8-inch spiral bound flip-cards. KDT scores were measured pre- and post-season using the same modality.

### 2.3. Analysis

For numerical measures, observations occurring outside of 1.5 times the interquartile range or appearing unlikely were evaluated for being outliers.

Nominal scores are summarized with mean and standard deviation to quantify the distribution of the responses under the normality assumption as well as median, first quantile, and third quantile to observe skewness (Table 2).

The test–retest reliability was quantified with the ICC (3, k), intra-class correlation, method described by Shrout and Fleiss (Table 3) [24,25]. ICC values < 0.5 are considered poor, 0.5 to 0.75 moderate, 0.75 to 0.9 good, and > 0.9 excellent [24,26]. All analyses were conducted using SAS v9.4 (SAS Institute; Cary, NC, USA).

## 3. Results

Following the rationale of Brooks and colleagues [15], we grouped the data by the following age bands: 5–7, 8–10, 11–12, and 13+ years old. Athletes < 13 years completed the SAC-C, while those who would be aged 13+ for most of the season completed the SAC. The pre-season baseline data for the sample are reported in Table 2. Age had a significant effect on all SAC/SAC-C subtests except Delayed Recall. SAC-C Orientation, Immediate Memory, and Concentration improved with age.

For the reliability analysis, 5–7-year-olds were combined with 8–10-year-olds due to small N in the 5–7 age band (Table 3). The pre- to post-season scores by the athletes and the best-fit lines are shown in Figure 2. The SAC/SAC-C total score displayed moderate test–retest reliability (ICC = 0.50 to 0.63) from the pre- to post-season, but with considerable variability in subtest reliability across the age ranges. Orientation displayed exceptionally poor reliability in the 5–10 (ICC = −0.14) and 13+ (ICC = 0.14) age bands, but moderate reliability (ICC = 0.62) in the 11–12-year-olds. Both Immediate Memory and Delayed Recall displayed poor reliability (ICC = −0.04 to 0.46) across all age bands. The Concentration subtest displayed moderate reliability (ICC = 0.50 to 0.71) across all age ranges, despite slight differences in item make-up and total score for the SAC-C and SAC versions of the task.

For mBESS, no errors were recorded for the DL stance for any athlete at baseline, whereas errors were typical in the TS and SL stance. The sample sizes were limited, but the adult version of the mBESS displayed moderate reliability (ICC = 0.65) in the 11–12-year-olds, but exceptionally poor reliability (ICC = −0.31) in the 13+ age group. mBESS-Child displayed poor reliability across all age bands.

An inspection of the TTG data led to all data from one administrator who consistently produced abnormally slow test times being removed (*n* = 38). Additional outlier analysis led to any TTG times < 9 or > 36 s (*n* = 12) being removed, regardless of the test administrator. After removing outliers, the pre-season TTG mean was 19.3 s (SD = 5.9; range 9.3–36). Age had a significant effect on TTG scores (*p* < 0.0001), with scores improving with age, but with wide ranges of performance. TTG displayed moderate to good stability from pre- to post-season testing across age bands (ICC = 0.67 to 0.81).

The overall mean baseline KDT time was 55.0 s (SD = 13.4; range 28.0–105.2). The tablet KDT administration produced slower mean test times than the flip-card administration [57.8 (14.2) vs. 50.8 (11.0), (*p* < 0.0001)], even after controlling for age; the KDT results are presented separately by administration mode. The KDT times improved with increasing age (*p* < 0.0001). The KDT demonstrated good to excellent test–retest reliability across age bands (ICC = 0.76 to 0.94) for both administration modes.

## 4. Discussion

This study, the largest youth dataset on the test–retest reliability of sideline concussion measures, illustrates the variable reliability and evolution of baseline performances as children age. Of the tests evaluated, oculomotor performance measured by the KDT demonstrated good to excellent reliability across all age ranges. The concentration subtest of the SAC/SAC-C demonstrated moderate test–retest reliability. The TTG demonstrated moderate to good reliability, while mBESS’s reliability was variable and typically poor.

The SAC is one of the most widely used sideline assessments of concussion despite its poor test–retest stability in large collegiate samples (ICC = 0.39) [9]. In our pediatric sample, the total score for SAC and SAC-C demonstrated moderate reliability with striking differences in reliability for individual subtests. Orientation showed poor (ages 5–10 and 13+) to moderate (11–12-year-olds only) reliability with broad confidence intervals. Pre-season testing was often completed during summer break and post-season testing during the school year. This timing difference could theoretically influence response accuracy for young athletes who may be more regularly reminded of the day and date during the regular school year than they are during summer break. Cognitive skills have been found to improve during the school year versus summer vacation; thus, post season improvement may be in part due to the active school year [27]. Future studies comparing pre- and post-season testing when both occur during the school year would be helpful to examine this observation further. Surprisingly, the immediate and delayed memory scores consistently showed poor reliability and were possibly influenced by restricted range, that is, only a 5-item list was used with many athletes performing near the top of the score range (i.e., ceiling effects). Concentration was the only SAC/SAC-C subtest that consistently demonstrated moderate reliability. The Concentration subtest primarily uses the well-established backward digit span task to assess working memory. Relative to other SAC/SAC-C subtests, Concentration showed notable between athlete variability (i.e., heterogeneity), while maintaining moderate within athlete stability, even in distracting settings. SAC’s limited reliability in the youth population is not significantly different from the collegiate population. SAC data from youth athletes should be assessed cautiously with limitations including age and perhaps season variation (summer vacation versus school year). It is clear that if the SAC is to be used for youth baseline testing, it should be completed annually to account for developmental improvement with age.

Balance assessment proved complicated. Our sample size was small, in part due to data from several athletes being excluded due to improbable results associated with differences in administration. After removing outliers and data from one rater who consistently produced outliers, the TTG achieved moderate to good stability in our sample. This is somewhat stronger than the poor stability (Pearson r = 0.46) reported by Nelson and colleagues [14]. TTG demonstrated an overall reduction in test time as age increased but was slower than previous reports using similar age bands [14,15,16], which also tended to differ from each other. Santo and colleagues [28] also found slightly longer TTG times in adolescents compared to our oldest athletes (17.16 s +/− 4.18 versus 16.7 s +/− 4.9), suggesting broad variability, even in large datasets. These observations suggest that SCAT3 TTG instructions alone may not be sufficient for appropriately standardized administration and acceptable inter-rater reliability.

For the mBESS, we included the SL stance and thus report scores for the age 13+ version of the mBESS across all age groups. No errors were recorded for DL stance for any athlete at baseline, whereas errors were typical in the performance of TL and SL stance. Nelson and colleagues [14] also included the adult version of the mBESS in their pediatric sample. Despite this, the total errors recorded in our sample were higher than those previously reported [14]; the gender distribution of the sample may be a contributing factor. Nelson and colleagues found that male participants committed significantly more errors than female participants on the mBESS, particularly the adult version [14]. While slightly more than 1/3 of the Nelson et al. sample was female, our sample was almost entirely male.

mBESS demonstrated moderate stability in 11–12-year-olds and exceptionally poor stability in the 13 and older age group. The mBESS-Child’s reliability was consistently poor. The differences in the mBESS test–retest reliability between ages has been previously reported. Prior work has found poor (Pearson r = 0.02) reliability for the mBESS in youth [14] and collegiate athletes (ICC = 0.41) [9]. We hypothesize several contributing factors. First, evaluating the ability of the examiner to understand the test instructions should be reviewed to alleviate this likely contributor to reduced reliability. Importantly, there are known effects of age and puberty on balance performance, with a non-linear developmental pattern expected. Studies using computerized forceplate technology have quantified these changes [29,30,31,32]. Young children (< 10 years) generally present with linearly defined, balanced performances which plateau between 11 and 13 years before developing again and reaching adult-like performance through approximately 14–16 years. These two plateaus correspond with expectations of sensorimotor processing that occur around these ages, with children < 10 years developing visual and somatosensory processes and appropriate integration of vestibular system information around 15 years. Based on the underlying physiological plateau in early adolescence and subsequent accelerated development, these age groups are especially prone to high variability in performance. Our findings were consistent with these expectations, but this does create challenges when attempting to establish reliable metrics for evaluating sideline injury. While baseline concussion screening, if completed, should be completed annually, it may be that some components, such as balance performance, may need more recent baselines to be effective as a monitoring tool. Another consideration regarding this sample relates to gender. We did not evaluate our sample in terms of gender due to the small number of females included, but it is established that boys within this age range generally perform more poorly on static balance measures than girls [32,33,34,35]. Finally, our test paradigm aimed at evaluating participants in distracting environments that reflect the reality of most recreational youth sport settings. Postural stability is associated with attention, and young children are known to have poor dual task balance performance, especially in vision-denied conditions (such as with mBESS) [36,37]. Ultimately, objective, quantitative tests of balance with superior reliability across ages and genders are required.

KDT testing demonstrated good to excellent stability pre- and post-season with evidence of improvement with increasing age, consistent with previous reports [38]. The KDT scores should improve through childhood and adolescence due to developmental changes in the brain and oculomotor pathways. Oculomotor coordination incorporates control from areas of the frontal lobe, which begin to stabilize around adolescence. Luna and colleagues also note that the developmental gains in oculomotor control are simultaneous with other developmental changes in the brain [39]. Improvement may also relate to other cognitive aspects affecting KDT performance including attention, language, and processing speed. While at least annual baseline KDT testing is essential for all, more frequent baselines (i.e., every 6 months) are recommended for athletes under 10, due to the wide range of scores and improvement in scores over time in youth athletes.

All measures demonstrated improvement with age and post-season testing, although age and/or summer vacation versus during the school year may have been contributing factors, the learning effect may have played a role. Repetitive test administrations can result in gains due to experience and learning of the task. A study looked at SAC and BESS scores from 50 athletes aged 9 to 14 for two test administrations, 60 days apart. There was a significant learning effect for BESS, but not SAC testing in this study [40]. In a youth athlete (13–18 years) study using the KDT, the learning effect was mild but present (43.9 versus 42.1 s) [41]. Test–retest reliability close to 0.9 are recommended for individual decision-making in a baseline assessment model [2]. The only sideline measure that consistently demonstrated good to excellent reliability in our pediatric sample was the KDT, with ICCs ranging from 0.83 to 0.94, consistent with data from collegiate samples [9,42]. Our data support recommendations by Echemendia [20] and Galetta [21] and colleagues to add a test of oculomotor function, such as the King–Devick Test, to the SCAT to improve sensitivity and sideline decision-making.

Our data lend support to the decision to remove Orientation, but retain Concentration in the SCAT5-C. The poor reliability demonstrated by the five-word list for immediate and delayed recall may provide theoretical justification for transition to use of the ten-word list. However, the reliability of the ten-word list in youth athletes has yet to be demonstrated. In the SCAT5, TTG was moved to the neurologic screen; only one trial was performed, it was no longer timed, and it was simply scored as normal or abnormal. Instructions are not further elaborated, nor is test surface specified. In our sample, we observed significant variability in timed performance. Brooks [15] noted a 2–6 s mean differences between their sample and the sample reported by Nelson [14] for comparable age groups and suggested that these differences may be due to “clinically important variability in testing instructions, timing, and environment” (p. 676) [15]. Given these observations, removal of the timing requirement seems reasonable. However, we would recommend use of an experienced examiner familiar with normal developmental variations in balance, as determining if a child’s balance is “normal” may prove challenging. Like mBESS, tandem gait is highly subjective, which increases room for variability, error, and inconsistent sideline decision-making. A consistent rater across the season is preferred but not easily achieved in recreational youth settings. The mBESS was retained in the SCAT5-C and SCAT6-C. Based on the exceptionally poor stability for the mBESS noted in our own and other [14] youth studies, it may not aid sideline or return-to-play decision-making for youth in its current form.

It is important to note that reliability is only one component of determining the validity and usefulness of a battery of tests. As outlined by Randolph and colleagues [2], validating a test battery for concussion management requires establishing test–retest reliability, sensitivity (i.e., ability to distinguish concussed athletes from non-concussed athletes), validity (i.e., does the test measure what it intends to measure), reliable change scores and classification rates (i.e., distinguishing normal variability from statistically significant change in performance—this relates to test–retest reliability), and clinical utility (i.e., clinical usefulness and practicality in the context in which it will be used). Reliable tests may not be sensitive to the effects of concussion and tests of modest reliability may still show sensitivity to concussion. Additional studies on the sensitivity and specificity of the SCAT components in youth are still needed.

This study has limitations. First, this study mostly evaluated male athletes and did not include sufficient females to evaluate gender effects. Female athletes may be particularly vulnerable to concussion and differences in recovery at all ages, and future studies will be required to determine the test–retest reliability of these measures over time in a pediatric female population. Second, the real-world setting could be considered a limitation or strength. Much of the pre- and post-season testing was conducted in distraction-filled locations near practice settings in outdoor parks, community buildings, or in available space near ice skating rinks. Surfaces and lighting available for balance testing varied (e.g., gym floor, sidewalk, low-pile carpet). Testing was often conducted before, during, or after practice or other team-related meetings. Third, test administrator experience varied from seasoned physician to research assistant. All test administrators were provided the standardized instructions for administering the SCAT components and KDT and were given the opportunity to observe, ask questions, and practice prior to administering the tests independently. While these factors almost certainly had a negative impact on test reliability, we believe that these settings reflect the reality of the facilities and personnel available for sideline/rink-side concussion testing in the “real world” of most youth sport practices and games—especially below the elite level. Fourth, while we attempted to recruit non-contact sport athletes to serve as an active control or comparison group, this proved challenging. We were unable to recruit enough non-contact sport athletes to analyze separately. Finally, as outlined in the Conflict of Interest section, a potential conflict of interest exists between some of the authors and King–Devick technologies. While the KDT was originally developed in the 1970s, during much of the conduct of this study, a co-branding agreement existed between King–Devick technologies and Mayo Clinic. Mayo Clinic employees do not directly benefit from this agreement or sales of the KDT. DWD has since joined the board of King–Devick technologies and holds options with the company. AT is a former employee of King–Devick technologies but was not involved in the design of the study or analysis of the data. JVW has a financial interest in Mayo Clinic Concussion Check, which incorporates the KDT.

Given the high rates of participation in sport, the high risk of concussion, and potential for long-term sequelae, youth athlete specific data and tools are necessary. Additional studies evaluating seasonal variation of cognitive testing during summer vacation versus when school is in session, the influence of environmental distractions, and the stages of puberty may provide additional guidance on the most appropriate parameters for the testing and interpretation of the measures. Tools that provide objective and quantitative results and are easy to use by any trained examiner are needed. More studies determining the feasibility of testing by laypersons would be useful. For example, the KDT screening tool has been shown to accurately identify concussed athletes when administered by non-medically trained individuals compared to healthcare professional [43,44].

## 5. Conclusions

It is feasible to complete baseline concussion testing in youth athletes, but overall test–retest reliability is not ideal for most screening tools, except the KDT. It is likely that there are numerous contributing factors, and further studies to better identify these factors may lead to the development of revised tools, new tools, or improved instruction and parameters necessary for the most accurate testing. Age is a significant contributing factor, and if baseline testing is to be completed with youth athletes, at least annual baseline testing is strongly recommended.

## Figures and Tables

**Figure 1 diagnostics-14-01661-f001:**
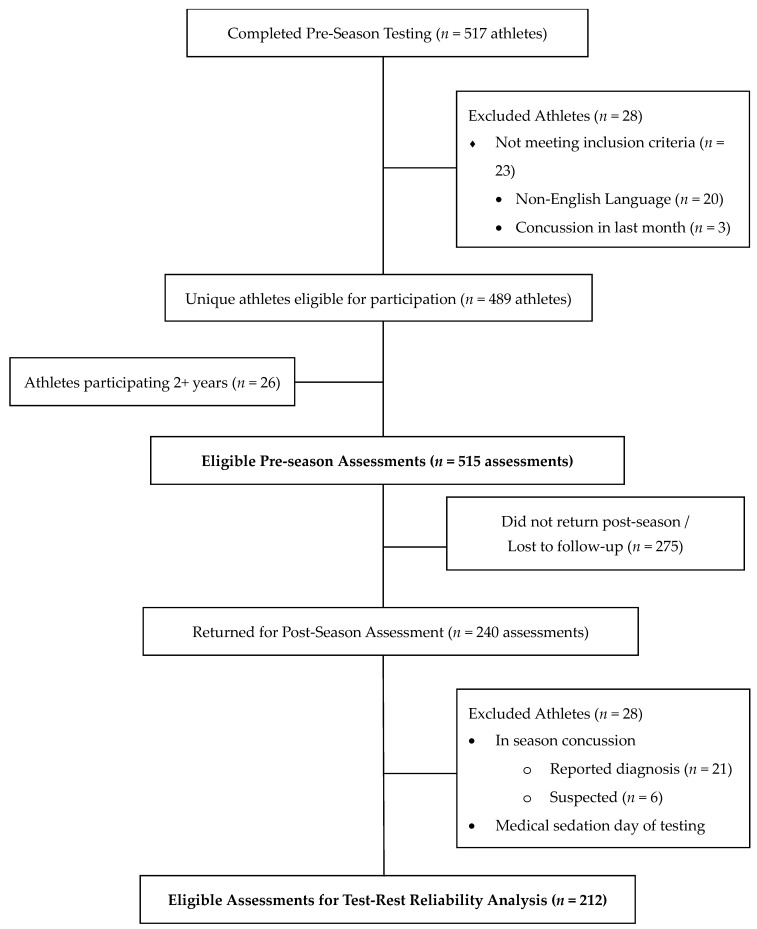
CONSORT Diagram.

**Figure 2 diagnostics-14-01661-f002:**
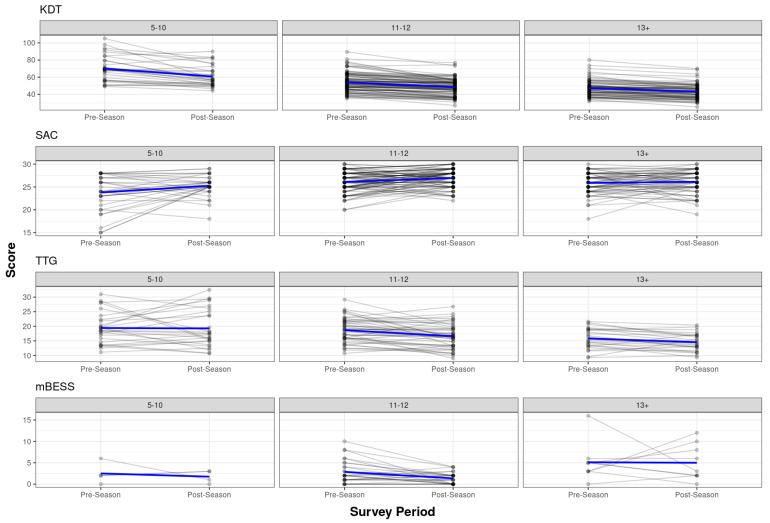
Pre- to post-season scores by athlete. Pre- to post-season test scores by test and age range. Individual lines represent individual athletes. Blue lines represent best fit.

**Table 1 diagnostics-14-01661-t001:** Demographics.

			Age (Years)		
	5–7 (*N* = 6)	8–10 (*N* = 112)	11–12 (*N* = 223)	13+ (*N* = 148)	Total (*N* = 489)
**Gender**, *n* (%)					
Female	0	13 (11.6%)	13 (5.8%)	6 (4.1%)	32 (6.5%)
Male	6 (100.0%)	99 (88.4%)	210 (94.2%)	142 (95.9%)	457 (93.5%)
**Primary Sport**, *n* (%)					
Missing	2	14	23	18	57
American Football—tackle	4 (100.0%)	82 (83.7%)	185 (92.5%)	90 (69.2%)	361 (83.6%)
American Football—non-tackle	0	2 (2.0%)	0	1 (0.8%)	3 (0.7%)
Ice Hockey	0	0	2 (1.0%)	34 (26.2%)	36 (8.3%)
Cheer	0	12 (12.2%)	13 (6.5%)	4 (3.1%)	29 (6.7%)
Baseball/Softball	0	0	0	1 (0.8%)	1 (0.2%)
Golf	0	1 (1.0%)	0	0	1 (0.2%)
Other	0	1 (1.0%)	0	0	1 (0.2%)
**Race/Ethnicity ***					
Am. Indian; Alaskan Native	0	0	1 (2.6%)	0	1 (0.9%)
Asian or Pacific Islander	0	1 (2.4%)	0	1 (3.6%)	2 (1.8%)
Black or African American	0	1 (2.4%)	3 (7.9%)	4 (14.3%)	8 (7.3%)
Hispanic or Latino	0	4 (9.8%)	3 (7.9%)	1 (3.6%)	8 (7.3%)
White or Caucasian	3 (100%)	29 (70.7%)	24 (63.2)	19 (67.9%)	75 (68.2%)
Multi-racial	0	5 (12.2%)	6 (15.8%)	3 (10.7%)	14 (12.7%)
Prefer not to answer	0	1 (2.4%)	1 (2.6%)	0	2 (1.8%)
Data unavailable *	3	71	185	120	379
**ADHD/Learning Disability ***					
ADHD	0	5/65 (7.7%)	8/113 (7.1%)	7/38 (18.4%)	20/219 (9.1%)
Learning Disability	0	2/65 (3.1%)	5/113 (4.4%)	4/39 (10.3%)	11/220 (5.0%)

* Race/Ethnicity and ADHD/Learning Disability data limitations: (1) In the initial years of the study, only questions included on the SCAT3/SCAT3-C form and those related to exclusion criteria were asked; and (2) When these demographic data were collected, they were collected via an optional online survey that was not completed by all parents.

**Table 2 diagnostics-14-01661-t002:** Normative and Reference Values.

	Age (Years)					
	5–7 (*N* = 6)	8–10 (*N* = 118)	11–12 (*N* = 238)	13+ (*N* = 153)	Total (*N* = 515)	*p*-Value
**SAC Orientation**						<0.0001 ^1,^*
N (Missing)	6 (0)	118 (0)	238 (0)	153 (0)	515 (0)	
Mean (SD)	2.3 (1.6)	3.2 (0.9)	3.6 (0.6)	4.7 (0.6)	3.9 (0.9)	
Median (IQR)	3 (1, 4)	3 (3, 4)	4 (3, 4)	5 (5, 5)	4 (3, 4)	
Range	0.0, 4.0	0.0, 4.0	2.0, 4.0	2.0, 5.0	0.0, 5.0	
**SAC Immed. Mem.**						<0.0001 ^1^
N (Missing)	6 (0)	118 (0)	238 (0)	152 (1)	514 (1)	
Mean (SD)	12.5 (1.6)	13.4 (1.5)	13.9 (1.2)	14.1 (1.1)	13.8 (1.3)	
Median (IQR)	13 (12, 13)	14 (13, 14)	14 (13, 15)	14 (14, 15)	14 (13, 15)	
Range	10.0, 15.0	5.0, 15.0	9.0, 15.0	11.0, 15.0	5.0, 15.0	
**SAC Concentration**						<0.0001 ^1,^*
N (Missing)	6 (0)	118 (0)	238 (0)	153 (0)	515 (0)	
Mean (SD)	2.5 (1.0)	3.7 (1.0)	4.1 (1.0)	2.8 (1.2)	3.6 (1.2)	
Median (IQR)	3 (2, 3)	4 (3, 4)	4 (3, 5)	3 (2, 4)	4 (3, 4)	
Range	1.0, 4.0	1.0, 6.0	1.0, 6.0	0.0, 5.0	0.0, 6.0	
**SAC Delayed Recall**						0.8624 ^1^
N (Missing)	6 (0)	117 (1)	238 (0)	153 (0)	514 (1)	
Mean (SD)	4.0 (0.9)	4.0 (1.3)	4.2 (0.9)	4.2 (0.8)	4.2 (1.0)	
Median (IQR)	4 (3, 5)	4 (4, 5)	4 (4, 5)	4 (4, 5)	4 (4, 5)	
Range	3.0, 5.0	0.0, 5.0	0.0, 5.0	2.0, 5.0	0.0, 5.0	
**SAC Total Score**						<0.0001 ^1^
N (Missing)	6 (0)	117 (1)	238 (0)	152 (1)	513 (2)	
Mean (SD)	21.3 (3.1)	24.4 (3.2)	25.8 (2.2)	25.9 (2.2)	25.5 (2.6)	
Median (IQR)	21 (19, 24)	25 (23, 27)	26 (24, 28)	26 (24, 28)	26 (24, 27)	
Range	18.0, 26.0	11.0, 28.0	19.0, 30.0	18.0, 30.0	11.0, 30.0	
**mBESS**						0.0384 ^1^
N (Missing)		6 (112)	74 (164)	25 (128)	105 (410)	
Mean (SD)		8.7 (5.0)	8.2 (6.0)	4.8 (4.0)	7.4 (5.7)	
Median (IQR)		9 (4, 12)	7 (3, 13)	4 (2, 6)	6 (3, 11)	
Range		3.0, 16.0	0.0, 20.0	0.0, 16.0	0.0, 20.0	
**mBESS-Child**						0.6281 ^1^
N (Missing)		6 (112)	75 (163)	25 (128)	106 (409)	
Mean (SD)		2.2 (2.0)	2.7 (2.9)	1.8 (2.0)	2.4 (2.7)	
Median (IQR)		2 (1, 2)	2 (0, 4)	1 (0, 3)	2 (0, 3)	
Range		0.0, 6.0	0.0, 10.0	0.0, 8.0	0.0, 10.0	
**Tandem Gait (TTG)**						<0.0001 ^1^
N (Missing)	5 (1)	103 (15)	135 (103)	85 (68)	328 (187)	
Mean (SD)	21.9 (4.0)	21.7 (6.6)	19.2 (5.2)	16.7 (4.9)	19.3 (5.9)	
Median (IQR)	21 (20, 23)	21 (16, 26)	19 (16, 23)	16 (13, 19)	19 (15, 23)	
Range	17.5, 28.2	10.3, 36.0	9.9, 33.0	9.3, 34.0	9.3, 36.0	
**KDT Flip-Card**						0.0006
N (Missing)	2 (0)	16 (0)	115 (0)	71 (0)	204 (0)	
Mean (SD)	74.0 (36.8)	57.5 (12.6)	51.6 (9.7)	47.3 (10.1)	50.8 (11.0)	
Median (IQR)	74 (48, 100)	58 (50, 67)	50 (45, 57)	46 (40, 54)	49 (43, 56)	
Range	48.0, 100.1	31.6, 74.6	33.8, 81.1	31.7, 80.1	31.6, 100.1	
**KDT Tablet**						<0.0001 ^1^
N (Missing)	4 (0)	102 (0)	122 (1)	82 (0)	310 (1)	
Mean (SD)	89.4 (20.0)	65.8 (14.3)	56.1 (11.5)	48.7 (9.0)	57.8 (14.2)	
Median (IQR)	96 (77, 102)	65 (56, 75)	55 (48, 64)	48 (42, 53)	56 (48, 66)	
Range	60.2, 105.2	29.4, 104.8	28.0, 89.5	31.3, 73.7	28.0, 105.2	

^1^ Kruskal–Wallis *p*-value. * SAC Orientation and Concentration, *p*-value comparison is for ages 5–12 only (e.g., SAC-C).

**Table 3 diagnostics-14-01661-t003:** Test–Retest Reliability Results.

	5–10 Years			11–12 Years			13+ Years			Total		
	*n*	ICC (95% CI)		*n*	ICC (95% CI)		*n*	ICC (95% CI)		*n*	ICC (95% CI)	
**KDT**	32	0.831	(0.653, 0.917)	101	0.922	(0.885, 0.948)	79	0.936	(0.899, 0.959)	212	0.929	(0.907, 0.946)
Flip-card	4	0.758	(−1, 0.984)	48	0.941	(0.894, 0.967)	41	0.93	(0.868, 0.962)	93	0.939	(0.908, 0.960)
Tablet	28	0.812	(0.593, 0.913)	53	0.89	(0.809, 0.937)	38	0.94	(0.885, 0.969)	119	0.912	(0.873, 0.938)
**SAC**	32	0.506	(−0.013, 0.759)	92	0.602	(0.398, 0.737)	66	0.632	(0.398, 0.774)	190	0.629	(0.505, 0.723)
Orient.	32	−0.139	(−1, 0.444)	92	0.623	(0.431, 0.751)	66	0.139	(−0.406, 0.473)	190	0.662	(0.549, 0.747)
Conc.	32	0.503	(−0.018, 0.757)	92	0.706	(0.556, 0.806)	66	0.705	(0.519, 0.82)	190	0.767	(0.689, 0.826)
Imm. Mem.	32	0.162	(−0.717, 0.591)	92	0.243	(−0.145, 0.499)	66	0.459	(0.116, 0.669)	190	0.377	(0.167, 0.533)
Delay R.	32	−0.036	(−1, 0.494)	92	0.444	(0.159, 0.632)	66	0.21	(−0.29, 0.516)	190	0.233	(−0.025, 0.426)
**TTG**	27	0.666	(0.268, 0.848)	45	0.753	(0.55, 0.864)	25	0.811	(0.572, 0.917)	97	0.749	(0.625, 0.832)
**mBESS**	2	-	-	14	0.651	(−0.088, 0.888)	9	−0.309	(−1, 0.705)	25	0.399	(−0.363, 0.735)
Child	4	0.075	(−1, 0.94)	28	0.356	(−0.392, 0.702)	9	0.144	(−1, 0.807)	41	0.287	(−0.337, 0.620)

Note: CI = Confidence Interval.

## Data Availability

The data presented in this study are available on request from the corresponding author.

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
