# Peer review of "Baseline Normative and Test–Retest Reliability Data for Sideline Concussion Assessment Measures in Youth"

_diagnostics, 2024, doi:10.3390/diagnostics14151661_

Round 1

Reviewer 1 Report

Comments and Suggestions for Authors

Mention that now there is SCAT 6, and with it some changes, including a 10 item word memory list.

What was the time period between pre and post season?  Was there a lot of variability in the range?  was it different by sport?  Time between pre and post test may impact practice effects.  

Did the sample have children with diagnosed learning disabilities, neurodevelopmental conditions (ADHD), etc?  add this info to the demographics. 

For the memory tasks, was there an alternate list provided?  In all cases or in some?  

Immediate and delayed memory may show poor ICC because with only 5 words, performance was near ceiling with not much variability. 

Comment in discussion about reliability not equaling validity – a reliable test may not be a sensitive test for concussion, that’s a separate study.  And poor retest reliability doesn’t exclude a measure from being sensitive to concussion.

Without accounting for normal age related changes over time, hard to know whether lack of a change pre versus post concussion is due to no cognitive deficit or a combination of developmental improvement but concussion related deficit.  I suggest publishing the means (SD) of your sample at baseline to offer this information to the larger community and to do so per ages separately (5, 6, 7, etc) versus age bands (8-10).  You have valuable information here that would be helpful normative data.

Reviewer 2 Report

Comments and Suggestions for Authors

The authors prospectively conducted pre-season and post-season testing of student athletes in football, ice hockey, and cheer to study baseline and test-retest reliability in commonly used concussion assessment tests.  515 players were included in the pre-season analysis, and 212 were included in the test-retest reliability analysis. The authors acknowledge the limitation of including very few females in the sample.  I agree that this is the main limitation.

Overall, this study was conducted well and written clearly.  Significant conflicts of interest exist between the authors and King-Devick test.  I have a couple recommendations to improve the manuscript.

1.       Please include the two most important sample size numbers in the abstract: 515 players in pre-season analysis, 212 in test-retest reliability analysis.

2.       Figure 1 has some partially cropped lines of text.

3.       The rightmost columns in Table 3 are cut off.

4.       It may be helpful to the reader to see a graphical representation of the pre to post-season means and SD or IQR, flowing from left (pre-season) to right (post-season).

Round 2

Reviewer 1 Report

Comments and Suggestions for Authors

No additional comments.

Author Response

Thank you for taking the time to review the manuscript. I do not see any specifics comments from Reviewer 2 to which I can respond. However, I believe the concern is addressed in the response to the Academic Editor. 

Sincerely, 

Jennifer Wethe, PhD, ABPP-CN